# Markers of kidney tubule dysfunction and injury and long-term risk of acute kidney injury following coronary artery bypass graft surgery

Lauren Shingler[1], Ashutosh Tamhane[2], Ching-Min Chu[1], Jennifer A. Frey[1], Emily B. Levitan[3], Suzanne E. Judd[3], Alexander L. Bullen[4], Edward D. Siew[5], Joseph V. Bonventre[6], Michael G. Shlipak[7], Byron Jaeger[8], Jesse C. Seegmiller[9], Alexander Keister[1], Orlando M. Gutierrez[2], Joachim H. Ix[4], Henry E. Wang[1]*

1 Department of Emergency Medicine, College of Medicine, The Ohio State University, Columbus, Ohio, United States of America, 2 Department of Medicine, Division of Nephrology, University of Alabama at Birmingham, Birmingham, Alabama, United States of America, 3 Department of Epidemiology, University of Alabama at Birmingham, Birmingham, Alabama, United States of America, 4 Department of Medicine, Division of Nephrology - Hypertension, University of California San Diego, UC San Diego Health, San Diego, California, United States of America, 5 Department of Medicine, Division of Nephrology and Hypertension, Vanderbilt University Medical Center, Nashville, Tennessee, United States of America, 6 Department of Medicine, Division of Renal Medicine, Brigham and Women's Hospital and Harvard Medical School, Boston, Massachusetts, United States of America, 7 Department of Nephrology, Kidney Health Research Collaborative, University of California San Francisco, and San Francisco Veterans Affairs Healthcare System, San Francisco, California, United States of America, 8 Bowman Gray Center for Medical Education, Wake Forest University School of Medicine, Winston-Salem, North Carolina, United States of America, 9 Department of Laboratory Medicine and Pathology, University of Minnesota, Minneapolis, Minnesota, United States of America

* Henry.wang@osumc.edu

**Data availability statement:** REGARDS data are not publicly available due to ethical and legal restrictions. To abide by its obligations with NIH/NINDS and the Institutional Review Board of the University of Alabama at Birmingham, REGARDS facilitates data sharing

## Abstract

### Background

Approximately one in five patients undergoing coronary artery bypass graft (CABG) surgery will develop post-operative acute kidney injury (AKI). We sought to determine whether biomarkers of kidney tubule dysfunction and injury are associated with long-term risk of acute kidney injury (AKI) after coronary artery bypass graft (CABG) surgery.

### Methods

We performed a cohort study using data from the REasons for Geographic And Racial Differences in Stroke (REGARDS), a national, population-based, longitudinal cohort study of 30,239 U.S. adults aged ≥45 years. Of 760 REGARDS participants who underwent CABG surgery after their REGARDS baseline visit, we excluded those with a history of dialysis or kidney transplant, missing laboratory data or with illegible or erroneous records, leaving 394 in the analysis. Exposures included urinary biomarkers of kidney tubule dysfunction (urine alpha-1 microglobulin [A1M], uromodulin [UMOD] and epidermal growth factor [EGF]) and injury (kidney injury

through data use agreements. Any investigator is welcome to access the REGARDS data, including statistical code, through this process. Requests for data access may be sent to regardsadmin@uab.edu.

**Funding:** This work was supported by award R01-DK128803 from the National Institute of Diabetes and Digestive and Kidney Diseases. The parent REGARDS study was supported by cooperative agreement U01-NS041588 from the National Institute of Neurological Disorders and Stroke (NINDS) and National Institute on Aging (NIA), National Institutes of Health (NIH), Department of Health and Human Services (DHHS). Additional funding was provided by R01-HL080477 and R01-HL165452 from the National Heart Lung and Blood Institute (NHLBI), and R37-DK39733, R01-DK072381 and U01-DK085660 provided by the National Institute of Diabetes and Digestive and Kidney Diseases. The content is solely the responsibility of the authors and does not necessarily represent the official views of the NINDS or the NIA. Representatives of the NINDS were involved in the review of the manuscript but were not directly involved in the collection, management, analysis or interpretation of the data. The authors thank the other investigators, the staff, and the participants of the REGARDS study for their valuable contributions. A full list of participating REGARDS investigators and institutions can be found at: REGARDS Investigators and Institutions.

**Competing interests:** The authors have declared that no competing interests exist.

**Abbreviations:** AKI: acute kidney injury; CKD: chronic kidney disease; ESRD: end-stage renal disease; A1M: alpha-1 microglobulin; UMOD: uromodulin; EGF: epidermal growth factor; KIM-1: kidney injury molecule-1; CABG: coronary artery bypass graft; CS: cohort study; SCr: serum creatinine

molecule-1 [KIM-1]), measured at an average of 5.5 years before CABG surgery. The primary outcome was AKI development following CABG surgery, defined as an increase in serum creatinine ≥0.3 mg/dL from 48 hours prior to CABG surgery to end of hospitalization.

## Results

Of 394 participants, 176 (45%) experienced post-CABG surgery AKI. Higher baseline urine A1M was associated with higher odds of AKI (adjusted OR 1.34 per 2-fold higher A1M, 95% Confidence Interval (CI): 1.00–1.80). Higher urine UMOD was associated with lower odds of AKI (adjusted OR 0.77 per 2-fold higher UMOD, 95% CI 0.62–0.95). Higher EGF showed a tendency towards lower odds of AKI (adjusted OR 0.79 per 2-fold higher EGF, 95% CI 0.59–1.05). KIM-1 was not associated with AKI (adjusted OR 0.92 per 2-fold higher KIM-1, 95% CI 0.77–1.10).

## Conclusions

Select biomarkers of kidney tubule dysfunction, but not injury, are associated with future risk of AKI after CABG surgery. Biomarkers measured years before surgery may predict post-CABG surgery AKI.

---

## Introduction

Acute kidney injury (AKI) is a major global public health problem, affecting 13−18 million patients and contributing to 1.5–1.7 million deaths worldwide each year [1]. In the United States alone, AKI accounts for $10 billion in additional healthcare expenditures per year [2].

In patients undergoing coronary artery bypass graft (CABG) surgery, AKI remains one of the most frequent and clinically important complications, affecting one in five patients post-operatively [3,4]. The pathophysiology of AKI following CABG surgery is multifactorial and incompletely understood. Potential reasons for AKI after CABG surgery include kidney ischemia from cardiopulmonary bypass, fluctuations in blood pressure and cardiac output, systemic inflammation, oxidative stress or underlying reduced kidney reserve [5]. Despite its high prevalence, it remains unclear why some patients develop post-CABG surgery AKI and others do not, despite having similar clinical profiles and surgical courses. Current clinical prediction models focus on immediate pre-operative risk factors but lack the ability to identify individuals at high AKI risk before the immediate pre-operative period. Early identification of risk factors may provide biological insights into mechanisms leading to AKI and may allow physicians to employ protective and preventive strategies to mitigate risk of AKI.

Conventional biomarkers of kidney health, such as estimated glomerular filtration rate (eGFR) and urine albumin-to-creatinine ratio (ACR), primarily reflect glomerular function and injury, but correlate poorly with degree of tubulointerstitial damage [6–8]. Given that kidney tubules are a major site of injury in AKI, assessment of kidney tubule health at baseline before the kidney stress from CABG may offer insights into AKI susceptibility beyond

what eGFR and ACR can detect [7]. Kidney tubule biomarkers include kidney injury molecule-1 (KIM-1), a marker of tubular injury; alpha-1 microglobulin (A1M), a marker of proximal tubule resorptive capacity; and uromodulin (UMOD) and epidermal growth factor (EGF), which reflect tubule protein synthetic capacity. These biomarkers have shown promise in quantifying susceptibility to poor kidney outcomes in the ambulatory setting such as CKD progression and cardiovascular disease (CVD) events, but have not been evaluated for future risk of AKI after CABG surgery beyond the immediate pre-operative setting [6,9–11].

We sought to determine the associations between kidney tubule dysfunction and injury markers with future risk of AKI following CABG surgery.

## Materials and methods

### Ethics approval and informed consent

The parent REasons for Geographic and Racial Disparities in Stroke (REGARDS) study and this ancillary study were approved by the Institutional Review Board of the University of Alabama at Birmingham. The parent REGARDS study obtained written consent from all cohort participants, including permission for review of medical records. The IRB waived requirement for informed consent for this ancillary study. The identities of the study participants were confidential and not accessible to the investigators.

### Study design and data collection – The REGARDS cohort

REGARDS is a longitudinal, population-based cohort including 30,239 community-living adults aged ≥45 years from across the continental US. Enrollment occurred between January 2003 and October 2007, and included a baseline interview where participants answered questions about demographics, health behaviors, diagnosed medical conditions and medication usage. As a part of an in-home visit, trained personnel collected vital sign measurements and obtained blood and urine samples from participants [12]. REGARDS conducted follow-up contacts with each participant at 6-month intervals for up to 19 years, identifying interim health and hospitalization events.

For this ancillary study, trained research personnel conducted structured chart reviews of hospitalizations for CABG surgery. Medical records included Emergency Department and admission notes, surgical notes, discharge summaries, and clinical laboratory values. Abstracted information pertinent to this study included history of maintenance dialysis, kidney transplant, use of dialysis during hospitalization and all creatinine values throughout the admission [13,14]. Research personnel had access to identifiable medical record information. The research team reviewed medical records during the period October 1, 2022 to May 1, 2024.

### Outcomes – Identification of Post-CABG surgery acute kidney injury

For the present study, we included participants who experienced hospitalizations associated with CABG surgery after their REGARDS baseline visit. The parent REGARDS study identified all hospitalizations and obtained copies of medical records for suspected coronary heart disease (CHD) events, which included CABG surgeries [12,15]. For participants who had multiple CABG surgery procedures during study follow-up, we selected the first surgery. We excluded participants who experienced a myocardial infarction or acute heart failure exacerbation during the CABG-associated hospitalization. Due to limitations in the available records, we were not able to obtain information on perioperative characteristics such as CABG procedure, bypass time, contrast exposure or medications.

The primary outcome of the study was the development of AKI in the post-operative period. We defined the participant's baseline creatinine as the lowest serum creatinine value in the 48 hours prior to CABG surgery and the post-CABG surgery creatinine as the highest creatinine value following CABG surgery. To define AKI presence and severity, we used the Kidney Disease Improving Global Outcomes (KDIGO) criteria, defined as a minimum rise in serum creatinine (SCr) of ≥0.3 mg/dL between the highest and baseline creatinine values [16].

### Exposures – Biomarkers of kidney tubule dysfunction and injury

The primary exposures were urine biomarkers of kidney tubule dysfunction (A1M, UMOD, EGF) and injury (KIM-1). These markers were based upon urine samples obtained near the time of participant enrollment in the parent REGARDS study. Blood and urine samples obtained during the initial in-home visit were centrifuged within 2 hours of collection, placed on ice and shipped overnight to the University of Vermont. Upon arrival, laboratory researchers centrifuged the samples at 30,000 $g$ and 4 °C.

The biomarker measurements for the current study were performed by Brigham and Women's Hospital. With the exception of A1M (which had a low CV), we measured all biomarkers twice and averaged the results to improve precision. We measured A1M using the SIEMENS BNII Nephelometer (Tarrytown, New York, USA) with a LLOD of 5.63 mg/L and intra-assay CV of 1.2–4.0%. We measured urine UMOD using the R-Plex Protocol on the Meso Scale Discovery (MSD) platform (Rockville, Maryland, USA) with a LLOD of 244.14 pg/mL and inter-assay CV of 6.4%. We measured KIM-1 and EGF using a microbead based ELISA assay from Bio-Rad Bio-Plex Luminex 200 reader (Hercules, California, USA). The lower limit of detection (LLOD) and inter-assay coefficients of variation (CV) for KIM-1 was 1.98 pg/mL and 7.8%, respectively. For EGF the LLOD was 0.313 pg/mL, and the CV was 9.2%.

### Covariates

Covariates included demographics (age, sex, race), urine creatinine (to account for urine tonicity at time of collection), body mass index (BMI), chronic medical conditions (hypertension and diabetes mellitus) and baseline glomerular kidney measures (eGFR and urine albumin) all measured at the REGARDS baseline visit. These covariates were selected for their biologic plausibility. We identified each chronic medical condition either through the participant's self-report or their use of disease-specific medications. Additionally, we included a diagnosis of diabetes mellitus if the participant had baseline laboratory values of either a fasting blood glucose of >126 mg/dL or a non-fasting blood glucose of >200 mg/dL. We calculated baseline eGFR using the participant's baseline serum creatinine and serum cystatin C measurements according to the 2021 CKD-EPI equation without race [17].

### Data analysis

We reported continuous variables using means and standard deviations or medians with quartiles, as appropriate. We examined the inter-relations between biomarkers using Spearman coefficients. We performed unconditional logistic regression (univariable and multivariable) to estimate the association between each biomarker (log$_2$ transformed and quartiles) with odds of AKI. The log$_2$ transformation accounts for data skew, allows interpretation as "per 2-fold higher", and facilitates comparison of strengths of associations with our prior work. We examined sequential multivariable models. Model 1 adjusted for age, sex, race, time from REGARDS interview to CABG surgery, and urine creatinine. Model 2 included Model 1 and history of diabetes, hypertension, and BMI at REGARDS enrollment. Finally, Model 3 included Model 2 variables and eGFR and urine albumin at enrollment. We also tested the associations between each biomarker and post-CABG surgery AKI stratified by time from REGARDS interview to CABG surgery hospitalization. Statistical significance was set at 0.05 (two-tailed), and we calculated 95% confidence intervals for all odds ratios. We performed all analyses using SAS version 9.4 (SAS Institute Inc., Cary, NC, USA).

### Results

Among the 30,239 REGARDS participants, 760 (2.5%) underwent CABG surgery [17]. We excluded 366 participants with illegible or incomplete records, 3 participants for a history of hemodialysis use or kidney transplant at hospital admission, 15 participants for missing serum creatinine values and 2 participants who underwent a cardiac procedure other than CABG, resulting in a final analytic sample of 394 individuals (Fig 1).

Of the 394 participants, 176 (45%) experienced AKI after their CABG surgery. Individuals who were male or Black were more likely to experience AKI. The AKI group had slightly lower eGFR values and higher urine ACR values at REGARDS baseline compared to the non-AKI group (Table 1).

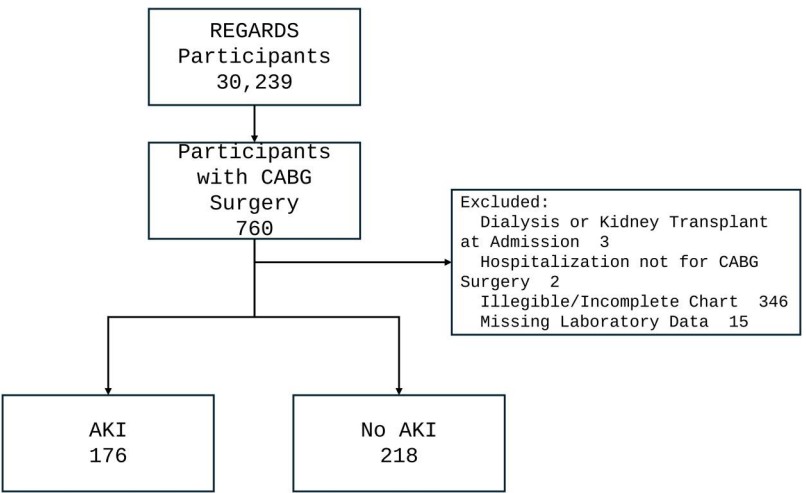

**Fig 1. Study population.** Sampling for this study within the population of participants admitted for CABG surgery in the REGARDS cohort. Abbreviations: AKI, acute kidney injury; CABG, coronary artery bypass graft; REGARDS, REasons for Geographic and Regional Differences in Stroke.

**Table 1. Baseline characteristics of participants at time of REGARDS enrollment.**

| Characteristic | AKI (n = 176) | No AKI (n = 218) | Total (n = 394) |
|---|---|---|---|
| **Demographics** | | | |
| Age, years – mean (SD) | 67 (7) | 65 (7) | 66 (7) |
| Sex, female – n (%) | 43 (24%) | 70 (32%) | 113 (29%) |
| Race, black – n (%) | 44 (25%) | 37 (17%) | 81 (20%) |
| **Health Behaviors** | | | |
| Body Mass Index, kg/m$^2$ – mean (SD) | 29 (5) | 29 (5) | 29 (5) |
| **Chronic Medical Conditions** | | | |
| Hypertension – n (%) | 123 (69%) | 126 (58%) | 249 (63%) |
| Diabetes – n (%) | 71 (40%) | 61 (28%) | 132 (33%) |
| **Baseline Glomerular Biomarkers** | | | |
| eGFR, ml/min/1.73 m$^2$ - mean (SD) | 78 (20) | 84 (17) | 81 (18) |
| Urine ACR mg/g – median (Q1, Q3) | 8.9 (5.3–24.6) | 7.6 (4.5–17.2) | 8.3 (4.8–20.3) |

Abbreviations: ACR, albumin to creatinine ratio; AKI, acute kidney injury; CABG, coronary artery bypass graft; eGFR, estimated glomerular filtration rate; SD, standard deviation; Q1, first quartile; Q3, third quartile

The kidney tubule markers showed weak to moderate correlation (Spearman correlation coefficient −0.12 to 0.50). (Table 2) The highest correlation (0.50) was between KIM-1 and EGF. Higher A1M was inversely correlated with eGFR. EGF and UMOD were directly correlated with eGFR, but the correlations were weak. Higher A1M was directly correlated with albuminuria, whereas EGFR and urine UMOD were inversely correlated with albuminuria.

Urine A1M values were higher in AKI than non-AKI participants. (Table 3) Conversely, urine UMOD and EGF values were lower in AKI than non-AKI participants. KIM-1 were similar in AKI vs. non-AKI participants. When stratified by quartiles, higher A1M was associated with higher odds of AKI, while higher urine UMOD and EGF quartiles were associated with lower odds of AKI. (Fig 2) Higher quartiles of KIM-1 were not associated with AKI development.

**Table 2. Spearman correlation coefficients between biomarkers of kidney tubule dysfunction and injury.**

|  | A1M/UCr | UMOD/UCr | EGF/UCr | KIM-1/UCr | eGFR | ACR |
|---|---|---|---|---|---|---|
| A1M/UCr | 1.0 | −0.07 | 0.05 | 0.26 | −0.12 | 0.25 |
| UMOD/UCr |  | 1.0 | −0.004 | −0.02 | 0.05 | −0.07 |
| EGF/UCr |  |  | 1.0 | 0.50 | 0.25 | −0.05 |
| KIM-1/UCr |  |  |  | 1.0 | −0.01 | 0.10 |
| eGFR |  |  |  |  | 1.0 | −0.01 |
| ACR |  |  |  |  |  | 1.0 |

Abbreviations: A1M, alpha-1 microglobulin; UMOD, uromodulin; EGF, epidermal growth factor; KIM-1, kidney injury molecule-1; UCr, urine creatinine; eGFR, estimated glomerular filtration rate; ACR, albumin:creatinine ratio

**Table 3. Biomarkers of kidney tubule dysfunction and injury stratified by CABG-Associated AKI.**

| Biomarkers | AKI<br>Median (Q1, Q3) | No AKI<br>Median (Q1, Q3) | Total<br>Median (Q1, Q3) |
|---|---|---|---|
| A1M, mg/L | 10.6 (6.3–16.9) | 7.8 (5.6–12.8) | 8.7 (5.7–15.1) |
| UMOD, µg/mL* | 8.5 (5.2–13.1) | 11.3 (6.6–17.4) | 10.0 (5.6–15.6) |
| EGF, ng/mL* | 6.6 (4.0–11.1) | 7.8 (4.9–12.5) | 7.3 (4.4–12.1) |
| KIM-1, ng/mL* | 1.0 (0.5–2.1) | 1.1 (0.5–1.8) | 1.1 (0.5–1.9) |

*Standard units are in pg/mL

Abbreviations: AKI, acute kidney injury; A1M, alpha-1 microglobulin; CABG, coronary artery bypass graft; EGF, epidermal growth factor; KIM-1, kidney injury molecule-1; Q1, first quartile; Q3, third quartile; UMOD, uromodulin

The median (interquartile range) time between REGARDS enrollment and CABG hospitalization was 5.5 years (2.7, 9.3); this did not differ between AKI (5.7 years [2.8, 9.2]) and non-AKI participants (5.5 years [2.7, 9.4]). sCr before CABG surgery was similar between AKI (median 1.0 mg/dL [0.8, 1.2]) and non-AKI participants (0.9 mg/dL [0.8, 1.1]). Following surgery, median sCr increase in those with AKI was 0.4 mg/dL [0.4, 0.7]) compared with those who did not have AKI (0.1 mg/dL [0.0, 0.2]).

Higher A1M was associated with higher odds of AKI following CABG surgery (adjusted OR per 2-fold higher A1M 1.34, 95% confidence interval (CI): 1.00–1.80). (Table 4) This association persisted with multivariable adjustment for potential confounders and with increasing quartiles of A1M. Higher UMOD and was associated with lower odds of post-CABG surgery AKI in all models. Higher UMOD quartiles were associated with lower odds of post-CABG surgery AKI, but this association did not reach statistical significance in the fully adjusted model. Higher EGF and EGF quartiles were associated with lower odds of post-CABG surgery AKI, but these relationships did not reach statistical significance in the fully adjusted model. Higher KIM-1 and KIM-1 quartiles were not associated with post-CABG surgery AKI.

When stratified by time to CABG surgery (<5 years vs. ≥5 years), there were no differences in associations between A1M and UMOD and future post-CABG surgery AKI. (Appendix 1) However, higher EGF was associated with lower odds of post-CABG surgery AKI in the <5 years but not the ≥5 years subset; <5 years OR 0.60 per 2-fold higher EGF (95% CI: 0.38–0.94), ≥5 years OR 1.00 per 2-fold higher EGF (95% CI: 0.67–1.50). Similarly, higher KIM-1 was associated with lower odds of post-CABG surgery AKI in the <5 years subset but not the ≥5 years subset; <5 years OR 0.66 per 2-fold higher KIM-1 (95% CI: 0.48–0.89), ≥5 years OR 1.12 per 2-fold higher KIM-1 (95% CI: 0.88–1.46).

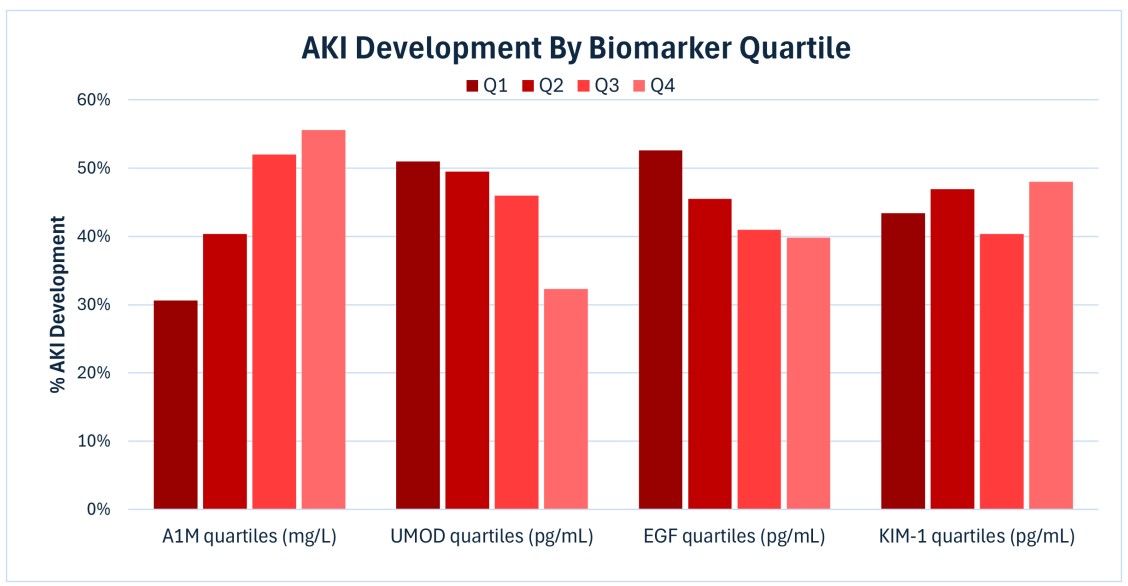

**Fig 2. AKI development stratified by biomarker measurement quartiles.** Abbreviations: AKI, acute kidney injury; A1M, alpha-1 microglobulin; EGF, epidermal growth factor; KIM-1, kidney injury molecule-1; Q1-Q4, first through fourth quartile; UMOD, uromodulin.

## Discussion

Our study builds upon our prior research from the SPRINT blood pressure trial, where we found that biomarkers of kidney tubular dysfunction (A1M, UMOD, EGF), but not injury, were associated with future development of AKI in a subgroup of CKD patients [6,18–21]. The current study suggests that markers of tubule dysfunction measured among community-based adults at a stable phase of health, may be associated with higher odds of AKI after future CABG surgery. These associations were independent of risk factors and glomerular kidney markers used routinely in clinical practice (eGFR and albuminuria).

In contrast to prior studies that evaluated CABG as the immediate stressor and cause of AKI, we evaluated kidney tubule health years earlier. Demirjian, et al. described a prediction model based on perioperative basic metabolic panel laboratory values with good predictive accuracy for moderate to severe acute kidney injury within 72 hours and 14 days after CABG surgery [22]. The TRIBE-AKI cohort showed that elevated pre-CABG surgery levels of sTNFR1, sTNFR2, and KIM-1 are associated with increased mortality, higher risk of cardiovascular events, and greater chance of CKD progression after CABG surgery, suggesting that subclinical injury and inflammation may have been present prior to the operation [23]. Our findings collectively build upon these prior studies, suggesting even earlier windows of opportunity for predicting individual vulnerability to post-CABG surgery AKI.

Prior studies largely focused on changes in KIM-1 in the perioperative period as a measure of dynamic tubular injury from the surgical procedure. This is different from the current study in which urine KIM-1 was measured in community-based individuals at a time of relative health, during which KIM-1's signal as a sensitive marker of acute tubular injury may have been limited. This observation is similar to other studies showing weak or absent associations of KIM-1 with outcomes such as rapid kidney function decline or incident hypertension in ambulatory populations in which it was measured during their healthy baseline [24]. Collectively, we infer from these data that KIM-1 may be most useful as a biomarker of acute tubular injury as opposed to a long-term marker of tubular injury.

The findings of this study provide new insights into mechanisms leading to AKI. Postulated mechanisms for AKI after CABG surgery include patient risk factors, intraoperative hemodynamic and inflammatory insults, and postoperative

**Table 4. Associations between biomarkers of tubule injury and dysfunction and post-CABG AKI.**

| | Unadjusted OR (95% CI) | Model 1 OR (95% CI) | Model 2 OR (95% CI) | Model 3 OR (95% CI) |
|---|---|---|---|---|
| **A1M mg/L** | | | | |
| Log$_2$(A1M) | 1.51 (1.20–1.89) | 1.47 (1.14–1.88) | 1.42 (1.09–1.84) | 1.34 (1.00–1.80) |
| Quartiles | | | | |
| Q1 | 1.00 | 1.00 | 1.00 | 1.00 |
| Q2 | 1.54 (0.85 - 2.77) | 1.50 (0.81 - 2.76) | 1.45 (0.78 - 2.70) | 1.34 (0.71 - 2.56) |
| Q3 | 2.46 (1.37–4.41) | 2.38 (1.26 - 4.52) | 2.24 (1.17 - 4.30) | 2.19 (1.10 - 4.36) |
| Q4 | 2.83 (1.58–5.08) | 2.70 (1.41 - 5.18) | 2.56 (1.32–4.98) | 2.24 (1.07 - 4.68) |
| **UMOD, pg/mL** | | | | |
| Log$_2$(UMOD) | 0.73 (0.60–0.89) | 0.70 (0.57–0.85) | 0.72 (0.59–0.88) | 0.77 (0.62–0.95) |
| Quartiles | | | | |
| Q1 | 1.00 | 1.00 | 1.00 | 1.00 |
| Q2 | 0.94 (0.54–1.65) | 0.85 (0.47 - 1.51) | 0.93 (0.52 - 1.69) | 0.96 (0.52 - 1.76) |
| Q3 | 0.82 (0.47–1.43) | 0.75 (0.42 - 1.34) | 0.82 (0.45 - 1.50) | 0.95 (0.51 - 1.77) |
| Q4 | 0.46 (0.26–0.82) | 0.42 (0.23 - 0.77) | 0.48 (0.26 - 0.90) | 0.60 (0.32 - 1.15) |
| **EGF, pg/mL** | | | | |
| Log$_2$(EGF) | 0.84 (0.70–1.01) | 0.70 (0.54–0.91) | 0.73 (0.56–0.95) | 0.79 (0.59–1.05) |
| Quartiles | | | | |
| Q1 | 1.00 | 1.00 | 1.00 | 1.00 |
| Q2 | 0.75 (0.43–1.32) | 0.59 (0.32 - 1.08) | 0.63 (0.34 - 1.17) | 0.68 (0.36 - 1.29) |
| Q3 | 0.63 (0.36–1.10) | 0.48 (0.26 - 0.91) | 0.51 (0.27 - 0.98) | 0.60 (0.30 - 1.20) |
| Q4 | 0.60 (0.34–1.05) | 0.39 (0.18 - 0.84) | 0.45 (0.21 - 0.98) | 0.61 (0.27 - 1.38) |
| **KIM-1, pg/mL** | | | | |
| Log$_2$(KIM-1) | 1.04 (0.92–1.18) | 1.01 (0.86–1.19) | 0.96 (0.81–1.14) | 0.92 (0.77–1.10) |
| Quartiles | | | | |
| Q1 | 1.00 | 1.00 | 1.00 | 1.00 |
| Q2 | 1.15 (0.66–2.02) | 1.01 (0.56 - 1.83) | 0.94 (0.51 - 1.73) | 0.88 (0.47 - 1.65) |
| Q3 | 0.89 (0.50–1.55) | 0.79 (0.42 - 1.49) | 0.69 (0.36 - 1.32) | 0.66 (0.34 - 1.31) |
| Q4 | 1.20 (0.69–2.10) | 1.09 (0.54–2.21) | 0.92 (0.45 - 1.90) | 0.76 (0.35 - 1.66) |

Model 1: Adjusted for age, sex (male and female), race (black and white), time from interview to CABG (<5 and ≥5 years) and urine creatinine.

Model 2: Model 1+adjusted for diabetes (yes and no), hypertension (yes and no) and BMI.

Model 3: Model 2+adjusted for CKD (eGFR<60 and >= 60) and log$_2$(urine albumin).

**Abbreviations:** AKI, acute kidney injury; A1M, alpha-1 microglobulin; UMOD, uromodulin; EGF, epidermal growth factor; KIM-1, kidney injury molecule-1; SCr, serum creatinine; eGFR, estimated glomerular filtration rate; OR, odds ratio; CI, confidence interval

complications [25]. The studied markers represent proximal tubule resorption (A1M) and distal synthetic dysfunction (EGF and UMOD) and reflect the functional integrity of the kidney. Globally, these findings suggest that subtle abnormalities in kidney tubule health may indicate a reduction in the kidney's resiliency or "kidney reserve" when responding to future stressors. In this context, CABG surgery may act as a type of kidney stress test where individuals with subtle and unrecognized kidney dysfunction may be more likely to develop AKI. The similarity of these findings to those of the SPRINT trial further support this hypothesis.

A potentially exciting implication of these findings is the ability to use urine biomarkers to identify high risk individuals vulnerable to post-surgical AKI. Early recognition of AKI risk could aid in shared decision making about the risks of the procedure and

any pre-emptive measures that might be taken beforehand, such as avoidance of contrast and nephrotoxic medications, pro-active hydration, or early admission for optimization of kidney health [25]. Among cardiac surgery patients identified as high-risk using urinary [TIMP-2]*[IGFBP7], the PrevAKI trial demonstrated that applying a KDIGO care bundle reduced the incidence and severity of post-operative AKI compared with standard care [26,27]. A similar approach might be applied using A1M, UMOD and EGF to identify high AKI-risk individuals, with prospective randomized assignment to AKI prevention interventions.

An important consideration is whether long-term biomarker degradation may have influenced the results of the study. Prior studies have examined shorter periods of time and have found small changes in stored sample UMOD, A1M, EGF and KIM-1 values without repeat freeze-thaw cycles [28–32]. During storage of the samples and analyses, the number of freeze-thaw cycles was minimized [27–30]. Typically, degradation of sample analytes would bias associations to the null. While not exhibiting significant associations in the full analysis, when stratified by elapsed time, EGF and KIM-1 exhibited associations with post-CABG surgery AKI in the<5-year subset. Replication with a larger series is the optimal strategy to assess the veracity of these findings.

Strengths of this study include the novel design and focus on characterizing kidney health years before AKI. The availability of multiple markers of kidney tubule health concurrently, and the detailed abstraction of daily creatinine values during the CABG surgery admissions are additional strengths. The study builds off hypotheses generated in another cohort with different etiologies of AKI, but provides similar findings, strengthening confidence in the inferences.

The study also had important limitations. The REGARDS cohort is limited to individuals ≥45 years; inclusion of younger participants may have amplified the observed associations by expanding the period of follow-up. We retrospectively abstracted clinical data through manual review of digital images of paper records. We could not differentiate the extent, types, methods, or urgency of CABG surgery. Factors other than CABG surgery may have triggered AKI. We did not have access to pre-hospitalization serum creatinine values prior to the CABG surgery admission. Missing medical records may have negatively biased the findings by obscuring AKI events.

## Conclusion

Biomarkers of tubule health, measured in community-based adults at a stable phase of health, are associated with AKI events after future CABG surgery. This finding was stronger for markers of kidney tubule dysfunction than for markers of tubule injury. Higher A1M and lower urine UMOD were associated with higher odds of AKI, independent of eGFR, albuminuria, and AKI risk factors. These findings provide new insights into the mechanisms of AKI after bypass surgery and provide opportunities to identify higher risk patients for post-CABG surgery AKI where preventive strategies may be warranted.

## Supporting information

**S1 Appendix. Associations between Biomarkers of Tubule Injury and Dysfunction and post-CABG AKI, stratified by time from REGARDS enrollment to CABG surgery. Models adjusted for age, sex (male and female), race (black and white), urine creatinine, diabetes (yes and no), hypertension (yes and no), BMI, CKD (eGFR<60 and ≥ 60) and log$_2$(urine albumin). Abbreviations: AKI, acute kidney injury; A1M, alpha-1 microglobulin; UMOD, uromodulin; EGF, epidermal growth factor; KIM-1, kidney injury molecule-1; SCr, serum creatinine; eGFR, estimated glomerular filtration rate; OR, odds ratio; CI, confidence interval.**
(DOCX)

## Acknowledgments

We would like to acknowledge the following individuals for contributions to medical record abstraction: Linda Cho, Ching-Min (Bryan) Chu, Derick Campbell, Eliane De Jong, Clemence Gatete, Cheryl Herchek, Emma Helkey, Emma Helwagen, Ireland Hester, Alexander Keister, Dina McGowan, Isabella Rigg, Riley Riggenbach, John Rider, Lauren Shingler, and Valerie Sircelj.

## Author contributions

**Conceptualization:** Lauren Shingler, Ashutosh Tamhane, Emily B. Levitan, Suzanne E. Judd, Alexander L. Bullen, Edward D. Siew, Joseph V. Bonventre, Michael G. Shlipak, Byron Jaeger, Jesse C. Seegmiller, Alexander Keister, Orlando M. Gutierrez, Joachim H. Ix, Henry E. Wang.

**Data curation:** Ashutosh Tamhane, Suzanne E. Judd, Alexander Keister.

**Formal analysis:** Ashutosh Tamhane, Byron Jaeger, Orlando M. Gutierrez, Joachim H. Ix, Henry E. Wang.

**Funding acquisition:** Orlando M. Gutierrez, Joachim H. Ix, Henry E. Wang.

**Investigation:** Lauren Shingler, Ashutosh Tamhane, Ching-Min Chu, Jennifer A. Frey, Joseph V. Bonventre, Michael G. Shlipak, Alexander Keister, Orlando M. Gutierrez, Joachim H. Ix, Henry E. Wang.

**Methodology:** Lauren Shingler, Ashutosh Tamhane, Emily B. Levitan, Alexander L. Bullen, Edward D. Siew, Byron Jaeger, Jesse C. Seegmiller, Orlando M. Gutierrez, Joachim H. Ix, Henry E. Wang.

**Project administration:** Jennifer A. Frey, Suzanne E. Judd, Alexander Keister, Orlando M. Gutierrez, Joachim H. Ix, Henry E. Wang.

**Resources:** Suzanne E. Judd, Joseph V. Bonventre, Michael G. Shlipak.

**Software:** Ashutosh Tamhane, Suzanne E. Judd.

**Supervision:** Ching-Min Chu, Jennifer A. Frey, Suzanne E. Judd, Orlando M. Gutierrez, Joachim H. Ix, Henry E. Wang.

**Visualization:** Lauren Shingler, Ashutosh Tamhane, Orlando M. Gutierrez, Joachim H. Ix, Henry E. Wang.

**Writing – original draft:** Lauren Shingler, Henry E. Wang.

**Writing – review & editing:** Lauren Shingler, Ashutosh Tamhane, Ching-Min Chu, Jennifer A. Frey, Emily B. Levitan, Suzanne E. Judd, Alexander L. Bullen, Edward D. Siew, Joseph V. Bonventre, Michael G. Shlipak, Byron Jaeger, Jesse C. Seegmiller, Alexander Keister, Orlando M. Gutierrez, Joachim H. Ix, Henry E. Wang.

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
