## [Decision Letter · Decision Letter 0]

26 Nov 2025

PONE-D-25-42789Kidney Tubule Dysfunction and Injury and the Long-Term Risk of Acute Kidney Injury Following Cardiac Artery Bypass Graft SurgeryPLOS ONE

Dear Dr. Wang,

Thank you for submitting your manuscript to PLOS ONE. After careful consideration, we feel that it has merit but does not fully meet PLOS ONE’s publication criteria as it currently stands. Therefore, we invite you to submit a revised version of the manuscript that addresses the points raised during the review process.

We look forward to receiving your revised manuscript.

Kind regards,

Ramada Rateb Khasawneh

Academic Editor

PLOS ONE

Journal Requirements:

Additional Editor Comments:

Abstract Needs Clearer Structure and Condensation

The abstract is overly detailed, especially the description of the parent cohort and exclusion criteria. This level of procedural detail belongs in the Methods section, not the abstract.

The objective and design are clear, but the results could be more concise. Avoid reporting redundant statistics (e.g., demographics already present in the Results section).

The conclusion should more explicitly highlight the clinical implications and novelty (i.e., biomarkers measured years before surgery predict AKI risk).

Clarity on Biomarker Categories

There is inconsistency in terminology: A1M and UMOD/EGF are labeled as “tubule dysfunction” markers, while KIM-1 is “injury.” It would help to consistently distinguish these throughout the paper and justify the categorization.

The narrative mixes up “higher quartiles associated with lower odds” and “associated with higher odds”—ensure clarity and correct logical direction.

Potential Confounders and Model Justification

Although multivariable models are described, it is unclear whether perioperative characteristics (e.g., bypass time, contrast exposure, pre-op medications) were unavailable or intentionally excluded. Their absence may lead to residual confounding.

Please justify why the time from REGARDS baseline to CABG (a median of 5.5 years) was treated as a covariate rather than stratified or included as an interaction term, since biomarker stability over time could influence effect estimates.

Interpretation of KIM-1 Null Findings

KIM-1 is widely considered a sensitive tubular injury marker. The manuscript should more clearly discuss why KIM-1 showed no association in this cohort, especially compared to TRIBE-AKI findings.

Possible explanations (biomarker degradation, baseline stability years before surgery, population differences) should be expanded.

Results Section Contains Errors and Ambiguity

There are mismatches in numbers and descriptions. For instance:

The statement regarding post-surgery creatinine change appears contradictory (“median 5.5 [4, 8] and 0.1 [0.0, 0.2]”)—this requires correction.

The sentence “higher A1M was associated with lower odds of AKI” contradicts the OR 1.34 (which indicates higher odds).

These inconsistencies reduce confidence in the reported findings.

Missing Discussion on Clinical Utility

The paper asserts that biomarkers may allow “early identification,” but the effect sizes are modest and CIs include the null (e.g., A1M lower bound 1.00). The discussion should temper conclusions and describe whether these biomarkers provide predictive value beyond existing models.

Strengths/Limitations Need Stronger Integration with Findings

The manuscript mentions biomarker degradation, restricted age range, and lack of pre-hospitalization creatinine—but does not explain how these factors may bias results.

Consider including a paragraph addressing the generalizability to younger populations and whether exclusion of 48% of CABG records (due to illegibility/incompleteness) may introduce selection bias.

Use consistent terms for CABG (either “CABG surgery” or “CABG”) throughout. Standardize “peri-CABG AKI,” “post-CABG AKI,” and “AKI following CABG.”

Reviewers' comments:

Reviewer's Responses to Questions

**Comments to the Author**

1. Is the manuscript technically sound, and do the data support the conclusions?

Reviewer #1: Yes

Reviewer #2: Yes

2. Has the statistical analysis been performed appropriately and rigorously?

Reviewer #1: Yes

Reviewer #2: I Don't Know

3. Have the authors made all data underlying the findings in their manuscript fully available?

Reviewer #1: Yes

Reviewer #2: No

4. Is the manuscript presented in an intelligible fashion and written in standard English?

Reviewer #1: Yes

Reviewer #2: Yes

5. Review Comments to the Author

Reviewer #1: I recommend the acceptance of this manuscript for publication. The study presents a novel and significant contribution to the field of nephrology and cardiac surgery by investigating the long-term predictive value of kidney tubular biomarkers for acute kidney injury (AKI) following coronary artery bypass graft (CABG) surgery. Its prospective design, rigorous methodology, and insightful findings offer a new paradigm for understanding AKI susceptibility, moving beyond immediate perioperative risk factors to assess renal health years in advance. The work is well-executed, clearly written, and has substantial potential to influence future clinical practice and research directions.

This study's core contribution lies in its innovative temporal approach, evaluating kidney tubular health at a stable baseline an average of 5.5 years before the stressor event, thus framing CABG as a "renal stress test" to uncover subclinical vulnerability. The research design is robust, nested within the large, population-based REGARDS cohort, and employs a well-defined cohort of 394 patients who underwent CABG. The methodology is commendable, utilizing precise measurements of specific biomarkers for tubular dysfunction (A1M, UMOD, EGF) and injury (KIM-1) and applying rigorous statistical models that adjust for a comprehensive set of confounders, including traditional markers like eGFR and albuminuria. The key findings are compelling: higher baseline A1M was associated with increased odds of AKI, while higher UMOD was associated with decreased odds, suggesting that markers of tubular dysfunction, rather than injury, are potent long-term predictors. Clinically, this opens a vital window for early risk stratification, paving the way for targeted preventive strategies to mitigate AKI risk in vulnerable patients undergoing cardiac surgery.

To further strengthen this already impactful manuscript, I offer the following specific suggestions for revision:

1.Enhance Clinical Context: In the discussion, please elaborate on the practical feasibility and potential cost-effectiveness of implementing these biomarker tests in routine pre-operative assessment pathways. A brief comparison of their predictive power against existing clinical risk models could also be valuable for clinicians.

2.Address Surgical Heterogeneity: While the study acknowledges limitations regarding surgical details, a brief discussion in the limitations section on how the lack of differentiation between elective and urgent cases, or variations in bypass time, might influence the observed associations would add nuance.

3.Clarify Biomarker Stability: The potential for biomarker degradation over long-term storage is mentioned as a limitation. It would be helpful to briefly cite any existing literature on the long-term stability of these specific urinary biomarkers under the storage conditions described, to better reassure the reader about the validity of the measurements.

4.Future Directions - Specificity: The suggestion for future research is excellent. To make it more concrete, consider proposing a specific follow-up study, such as a prospective trial where high-risk patients identified by these biomarkers are randomized to a KDIGO-guided preventive bundle versus standard care to assess outcomes.

5.Additional discussion: The author can further explore in the discussion section the detectability, cost-effectiveness of markers such as A1M in clinical practice, as well as their potential for integration with existing clinical risk prediction models (such as the Demirjian model).

Reviewer #2: The manuscript is an adequate scientifc research. It is written in standard English. Requirement of informed consent has been waived. however, regarding data availabilty, some restrictions has been applied

6. PLOS authors have the option to publish the peer review history of their article (what does this mean?). If published, this will include your full peer review and any attached files.

Reviewer #1: No

Reviewer #2: No

---

## [Author Response · Author response to Decision Letter 1]

9 Feb 2026

Full responses to reviewers have been attached as a separate document.

---

## [Decision Letter · Decision Letter 1]

26 Mar 2026

PONE-D-25-42789R1Markers of kidney tubule dysfunction and injury and long-term risk of acute kidney injury following cardiac artery bypass graft surgeryPLOS One

Dear Dr. Wang,

Thank you for submitting your manuscript to PLOS ONE. After careful consideration, we feel that it has merit but does not fully meet PLOS ONE’s publication criteria as it currently stands. Therefore, we invite you to submit a revised version of the manuscript that addresses the points raised during the review process.

Please address the reviewer comments

We look forward to receiving your revised manuscript.

Kind regards,

Ramada Rateb Khasawneh

Academic Editor

PLOS One

Journal Requirements:

Additional Editor Comments:

Please address the reviewer comments

Reviewers' comments:

Reviewer's Responses to Questions

**Comments to the Author**

1. If the authors have adequately addressed your comments raised in a previous round of review and you feel that this manuscript is now acceptable for publication, you may indicate that here to bypass the “Comments to the Author” section, enter your conflict of interest statement in the “Confidential to Editor” section, and submit your "Accept" recommendation.

Reviewer #3: All comments have been addressed

Reviewer #4: All comments have been addressed

2. Is the manuscript technically sound, and do the data support the conclusions?

Reviewer #3: Yes

Reviewer #4: Yes

3. Has the statistical analysis been performed appropriately and rigorously?

Reviewer #3: Yes

Reviewer #4: I Don't Know

4. Have the authors made all data underlying the findings in their manuscript fully available?

Reviewer #3: Yes

Reviewer #4: Yes

5. Is the manuscript presented in an intelligible fashion and written in standard English?

Reviewer #3: No

Reviewer #4: Yes

6. Review Comments to the Author

Reviewer #3: The authors have tried to respond to many questions. But some of the sections have overlapped to each other.

Reviewer #4: Thank you for the revision. The manuscript is improved and most substantive concerns have been addressed. However, the clean revised version still contains several obvious internal errors that should be corrected before acceptance. The main remaining issues are: (1) the title uses “cardiac artery bypass graft” instead of “coronary artery bypass graft”; (2) the cohort size is given as 30,239 in the manuscript background but 30,329 in the Results; (3) the Figure 1 legend refers to the “REGARDS Trial” rather than the REGARDS cohort/study; (4) Table 3 appears to contain an impossible EGF summary value for the AKI group (6.6 [4.0–1.1]); (5) at least one Results sentence remains garbled (“lower higher odds,” “while whereas”); and (6) the PrevAKI trial is described as ongoing even though the cited study is a published randomized trial. These look fixable and do not seem to undermine the overall scientific message, but they reflect insufficient final quality control. I would therefore recommend revision rather than acceptance in the current form.

7. PLOS authors have the option to publish the peer review history of their article (what does this mean?). If published, this will include your full peer review and any attached files.

Reviewer #3: No

Reviewer #4: No

---

## [Decision Letter · Decision Letter 2]

29 Apr 2026

Markers of kidney tubule dysfunction and injury and long-term risk of acute kidney injury following coronary artery bypass graft surgery

PONE-D-25-42789R2

Dear Dr. Wang,

We’re pleased to inform you that your manuscript has been judged scientifically suitable for publication and will be formally accepted for publication once it meets all outstanding technical requirements.

Kind regards,

Ramada Rateb Khasawneh

Academic Editor

PLOS One

Additional Editor Comments (optional):

Good Luck

Reviewers' comments:

Reviewer's Responses to Questions

**Comments to the Author**

1. If the authors have adequately addressed your comments raised in a previous round of review and you feel that this manuscript is now acceptable for publication, you may indicate that here to bypass the “Comments to the Author” section, enter your conflict of interest statement in the “Confidential to Editor” section, and submit your "Accept" recommendation.

Reviewer #3: All comments have been addressed

Reviewer #4: All comments have been addressed

2. Is the manuscript technically sound, and do the data support the conclusions?

Reviewer #3: Yes

Reviewer #4: Yes

3. Has the statistical analysis been performed appropriately and rigorously?

Reviewer #3: Yes

Reviewer #4: Yes

4. Have the authors made all data underlying the findings in their manuscript fully available?

Reviewer #3: Yes

Reviewer #4: Yes

5. Is the manuscript presented in an intelligible fashion and written in standard English?

Reviewer #3: Yes

Reviewer #4: Yes

6. Review Comments to the Author

Reviewer #3: He has tried to respond to most of the questions but please revise on the Authors contribution still not completed with some of the authors missing and some categories have no specific authors.

Reviewer #4: (No Response)

7. PLOS authors have the option to publish the peer review history of their article (what does this mean?). If published, this will include your full peer review and any attached files.

Reviewer #3: No

Reviewer #4: No

---

## [Editor Report · Acceptance letter]

PONE-D-25-42789R2

PLOS One

Dear Dr. Wang,

I'm pleased to inform you that your manuscript has been deemed suitable for publication in PLOS One. Congratulations! Your manuscript is now being handed over to our production team.

Kind regards,

on behalf of

Dr. Ramada Rateb Khasawneh

Academic Editor

PLOS One